# High-Grade Gliomas in Children—A Multi-Institutional Polish Study

**DOI:** 10.3390/cancers13092062

**Published:** 2021-04-24

**Authors:** Aleksandra Napieralska, Aleksandra Krzywon, Agnieszka Mizia-Malarz, Joanna Sosna-Zielińska, Ewa Pawłowska, Małgorzata A. Krawczyk, Katarzyna Konat-Bąska, Aneta Kaczorowska, Anna Dąbrowska, Maciej Harat

**Affiliations:** 1Radiotherapy Department, Maria Sklodowska-Curie National Research Institute of Oncology Gliwice Branch, 44-101 Gliwice, Poland; 2Department of Biostatistics and Bioinformatics, Maria Sklodowska-Curie National Research Institute of Oncology, Gliwice Branch, 44-101 Gliwice, Poland; aleksandra.krzywon@io.gliwice.pl; 3Department of Pediatrics, Medical University of Silesia, 40-752 Katowice, Poland; amizia-malarz@sum.edu.pl (A.M.-M.); mlodasosna@poczta.onet.pl (J.S.-Z.); 4Department of Oncology and Radiotherapy, Faculty of Medicine, Medical University of Gdansk, 80-210 Gdansk, Poland; ewa.pawlowska@gumed.edu.pl; 5Department of Pediatrics, Hematology and Oncology, Medical University of Gdansk, 80-210 Gdansk, Poland; mkrawczyk@gumed.edu.pl; 6Wroclaw Comprehensive Cancer Center, 53-413 Wrocław, Poland; konat.katarzyna@dco.com.pl; 7Department of Oncology, Wroclaw Medical University, 53-413 Wrocław, Poland; 8Department of Children Oncology and Haematology, Wroclaw Medical University, 53-413 Wrocław, Poland; kaczorowskaa@usk.wroc.pl; 9Department of Paediatrics, Haematology and Oncology Nicolaus Copernicus University in Toruń Collegium Medicum in Bydgoszcz, 85-094 Bydgoszcz, Poland; anna_dabrowska83@wp.pl; 10Department of Neurooncology and Radiosurgery, Franciszek Lukaszczyk Memorial Oncology Center, 85-796 Bydgoszcz, Poland; haratm@co.bydgoszcz.pl

**Keywords:** high-grade glioma, pediatric oncology, radiotherapy, chemotherapy, glioblastoma multiforme, anaplastic astrocytoma

## Abstract

**Simple Summary:**

High-grade gliomas constitute less than 5% of pediatric brain tumors. Due to the rarity of such a diagnosis, the lack of consensus about the best therapeutic approach, and the difficulty in conducting prospective trials; a retrospective multi-institutional analysis, such as the one presented in this article, is needed. We carried out the survival analysis of children diagnosed and treated with high-grade gliomas in seven major polish institutions. The assessment of the outcome of 82 consecutive patients with grade III and grade IV tumors was performed and showed a 5-year overall survival of only 30%. The extent of resection, immediate temozolomide-based chemotherapy, and radical radiotherapy were found as factors positively influencing survival.

**Abstract:**

Due to the rarity of high-grade gliomas (HGG) in children, data on this topic are scarce. The study aimed to investigate the long-term results of treatment of children with HGG and to identify factors related to better survival. We performed a retrospective analysis of patients treated for HGG who had the main tumor located outside the brainstem. The evaluation of factors that correlated with better survival was performed with the Cox proportional-hazard model. Survival was estimated with the Kaplan–Meier method. The study group consisted of 82 consecutive patients. All of them underwent surgery as primary treatment. Chemotherapy was applied in 93% of children with one third treated with temozolomide. After or during the systemic treatment, 79% of them received radiotherapy with a median dose of 54 Gy. Median follow-up was 122 months, and during that time, 59 patients died. One-, 2-, 5-, and 10-year overall survival was 78%, 48%, 30% and 17%, respectively. Patients with radical (R0) resection and temozolomide-based chemotherapy had better overall survival. Progression-free survival was better in patients after R0 resection and radical radiotherapy. The best outcome in HGG patients was observed in patients after R0 resection with immediate postoperative temozolomide-based chemotherapy and radical radiotherapy.

## 1. Introduction

Glioblastoma multiforme (GBM), anaplastic astrocytoma (AA), and anaplastic oligodendroglioma (AO) are the most common high-grade gliomas (HGG) diagnosed in the adult population. In pediatric patients, HGG are much rarer and constitute less than 5% of brain tumors [1,2]. The 5-year overall survival (OS) of adult patients with grade III tumors is less than 50%, while less than 10% of grade IV patients live more than 5 years [2]. Childhood central nervous system cancers, while rare, contribute substantially to cancer-related mortality in this population, surpassing other cancers as the main reason for cancer mortality [2,3]. While in adult patients, treatment approach is well established and based on the tumor mutation profile and age of the patient, treatment in children varies between institutions, and there is lack of consensus about the best therapeutic schema. To date, maximal safe resection combined with chemotherapy (CTH) and local radiotherapy (RT) is the mainstay of treatment, even though the timing of the irradiation and the choice of systemic agent is still a matter of debate [4,5]. With only one prospective trial and few retrospective studies proving the benefit of temozolomide-based CTH, the value of this therapeutic agent is still a subject of discussion [6,7,8,9,10]. The use of adjuvant CTH after RT or radiochemotherapy (RCTH) and the sequence of treatment or total dose of RT also varied between the studies [1,6,7,8,9,10,11,12,13,14,15,16,17,18,19,20,21,22,23,24,25,26,27,28,29,30,31]. The above-mentioned, together with the rarity of HGG diagnosis in the pediatric population, strongly calls for a multi-institutional analysis, such as the one presented in this article [32,33,34,35].

The aim of our study is to assess the long-term results of treatment of children with HGG in the last 25 years and to identify factors related to better survival. Hence, we report the results of the multi-institutional cohort analysis.

## 2. Materials and Methods

A retrospective analysis of children with primary HGG (GBM, AA, and AO), treated in seven major institutions (two hospitals in Bydgoszcz, one in Gdańsk, one in Gliwice, one in Katowice, and two in Wrocław) in Poland between 1994 and 2019, was performed. Cooperation between RT centers and children oncologic departments was established in order to conduct this study. We included all consecutive patients younger than 18 years old who were diagnosed with HHG during the study period, had main tumor located outside the brainstem, and received at least 2 months of follow-up from the date of diagnosis. In all the cases, the diagnosis was based on diagnostic imaging (computed tomography (CT) and, in later years of the study, magnetic resonance, (MR)) and pathologic examination of the tumor tissue samples obtained during a biopsy or surgery. The presence of previous neoplasm was not an exclusion criterion. Medical records were screened individually in each participating center and all of the records were verified by the first author. The study was approved by the ethical committee in MSC National Research Institute of Oncology in Gliwice (number: KB/430-05/20) and performed according to the Helsinki Declaration.

Overall survival (OS) was calculated from the date of diagnosis to the date of death or last follow-up. Progression-free survival (PFS) was calculated from the date of diagnosis to the date of progression of the tumor or death of the patient. Eastern Cooperative Oncology Group (ECOG) scale was used to classify patients’ performance status during RT. Progression was assessed with diagnostic imaging unless observed clinically apparent.

Parameters included in analysis were: age, tumor histopathological grade, the date of primary diagnosis, given curative treatment, patients’ performance status, type of CTH, extent of resection, total dose of RT, disease symptoms, tumor location (supra/infratentorial), time between the surgery and RT, time between the surgery and CTH, response to RT, the date of progression, and death. Missing dates of deaths were obtained from the Polish National Cancer Registry. All the various patients’ characteristics were included in the univariate and multivariate analysis to identify their impact on OS and PFS.

In the statistical analysis, OS and PFS were estimated using the Kaplan–Meier method. Median follow-up was estimated by the Kaplan–Meier analysis with the reversed meaning of the status indicator. Comparisons were made with the use of the log-rank test. The Cox proportional-hazards model for the univariate and multivariate analyses of the prognostic factors was applied. Variables with *p*-value of <0.05 in the univariate Cox analysis were used in the multivariate Cox analysis. The Fisher’s exact test was used to determine the association between EOR and grade. Two-sided *p* values of <0.05 were considered statistically significant. Statistical analysis was performed using Statistica software (version 13.3.721.1, Stasoft Polska TIBICO Software Inc. Palo Alto, CA, USA) and R statistical software package version 4.0.1. released in June 2020 (R Foundation for Statistical Computing, Vienna, Austria, http://www.r-project.org).

## 3. Results

### 3.1. Patients Characteristics

The study group consisted of 82 consecutive patients. The patients’ and treatment characteristics are presented in Table 1. Family history of brain tumor was positive in the cases of five patients. Eight patients had the history of previous oncologic treatment—three were previously diagnosed with astrocytoma grade II, two with rhabdomyosarcoma, two with acute lymphoblastic leukemia (ALL), and one patient was treated for retinoblastoma a few years earlier. The majority of patients had the diagnosis based on MR, except for those treated in early years of the study. In one patient, the tumor was bifocal; in another, dissemination to meninges was diagnosed; and all other patients were diagnosed with a single brain tumor. Mean tumor dimensions were 48.6 × 44.5 × 44.8 mm.

The most common symptoms were headaches (47.7%) and vomiting (41.5%). The mean and median time of duration of the symptoms was 81 and 30 days, respectively. In one patient, tumor was diagnosed in prenatal examination; one patient had diagnostic imaging during routine ALL follow-up; and in two patients, head injury was the reason to perform diagnostic imaging. All those children were lacking any symptoms.

### 3.2. Primary Treatment

All patients had surgery as the first step of primary treatment. Only 14 patients had radical (R0) resection confirmed with MR. Among them, six had Grade III tumors and eight had Grade IV tumors. The second and third surgery was performed in 18 and 2 patients, respectively. CTH was applied in 93% of children. In Poland, the majority of pediatric oncologic centers applied CTH according to national protocols. However, varied treatment protocols were employed due to the long study period. At least 40% received one (usually four) cycle of irinotecan plus carboplatin, 30% received systemic treatment based on temozolomide +/− cisplatin, and 39% received ifosfamid with adriamycin or vincristine. Among patients who received temozodomide-based CTH, 12 had WHO G III tumors and 13 had WHO GIV tumors. In nine patients, precise information about the systemic treatment was not available. The time from surgery to the beginning of CTH ranged from 0 to 153 days with mean and median of 30 and 22 days, respectively. Some children received combined treatment (RT with temozolomide), especially in the latter years of the study. The response to systemic treatment was assessed in 68 patients, with complete regression observed in one patient, partial regression in nine, stable disease in 17 and progression of the lesion in 41 patients.

After or during systemic treatment, 65 (79%) children received RT. The time between the first surgery and the beginning of RT ranged from 12 to 511 days (median and mean of 132 and 143, respectively). Mean and median time to the start of irradiation was usually longer in children younger than 5 years old (mean and median of 259 and 178, respectively) compared to the older ones (mean and median of 128 and 132, respectively). The difference in delay of the offset of RT was the most pronounced in the group of children younger than 3 years old, with mean and median of 341 and 398 days, respectively. Radical RT was applied in 59 patients and palliative in six. Four patients did not complete RT due to the deterioration of performance status and finished irradiation after total dose of 3.6 Gy, 12 Gy, 37.8 Gy, and 46.8 Gy, respectively.

Total dose used in radical and palliative RT ranged from 46.8 to 70 Gy (median 54, mean 55.9) and from 3.6 to 39.6 Gy (median 30, mean 25.5), respectively. In all of the patients, treatment planning was based on 3D imaging. A thermoplastic mask was used for each patient for treatment position reproducibility. A CT scan was done in the treatment position for treatment planning. CT (and MR images in the later years of the study) were used to define target volumes and normal structures. The gross tumor volume (GTV) included the operative bed and residual tumor as defined by contrast-enhanced T1-weighted, T2-weighted, and fluid attenuated inversion recovery (FLAIR) sequences. In pre-MR era, preoperative and postoperative contrast-enhanced tumor and tumor bed visible on CT was considered GTV. The GTV was expanded by an anatomically constrained margin from the clinical target volume (CTV). An additional geometric expansion was added to the CTV to create the planning target volume (PTV). The PTV was meant to account for set-up errors and intrafraction motion. Two-stage RT was applied in 23 children, one-stage in 42. In all of the patients, after non-radical surgery in the first stage, residual tumor +/− tumor bed was irradiated, but in five patients, due to the extent of the disease, the irradiation field was larger and also covered the ventricular system (one patient), cerebellum and brainstem (one patient), craniospinal axis (one patient), middle cranial fossa (one patient), or whole brain (one patient). In patients, after radical resection, tumor bed with margin was irradiated. In the second stage of RT (boost field, total dose ranged from 5.4 to 18 Gy) tumor bed +/− residual tumor was irradiated unless total resection was performed. Total dose over 54 Gy was used in 24 patients. Mean GTV, CTV, and PTV was 76 ± 83 mm^2^, 239 ± 244 cm^2^, and 336 ± 280 mm^2^, respectively. Margin between GTV/CTV and PTV ranged from 5 to 45 mm (mean 20). RT was delivered with the use of 6–23 MV X photon beams. Response to RT was assessed in 16 patients with good response (partial or complete regression) observed in 14 of them.

### 3.3. Recurrence of the Disease

Progression of the disease was observed in 43 patients with median PFS of 122 months. ECOG performance status at the time of recurrence diagnosis of 0, 1, or 2 and more was observed in 4, 21, and 18 patients, respectively. In 12 patients, ECOG at the time of recurrence diagnosis was worse than at the time of primary tumor diagnosis. A recurrence location description was available in 34 out of 43 patients, and tumor bed was the most common place of disease progression (in 31 patients), with only three cases of the progression outside the previous lesion location. Metastases were diagnosed in 10 patients during the follow-up.

Twenty-eight (65%) among 43 patients diagnosed with recurrence received treatment. A deterioration in the performance status was the most common reason for therapy discontinuation. Surgery, RT, and CTH were implemented in 18, 14, and 23 patients, respectively. Among those patients who received irradiation, the conventional RT was used in nine patients (in one combined with stereotactic boost of 5 Gy), the stereotactic RT in six, and in one case, the RT description was unavailable. Total and fraction dose used in conventional irradiation ranged from 16.2 to 60 Gy (median 54) and 1.8 to 3 Gy (median 1.8), respectively. In three patients, it was the second course of RT. Stereotactic RT was delivered with the median fraction dose of 8 Gy (range 5 to 12) to median total dose of 12 Gy (range 5 to 16). In five patients, stereotactic treatment was the second course of RT, and in two of them, it was the third. Only one patient had the deterioration of performance status during RT and finished irradiation at the dose of 16.8 Gy.

### 3.4. Survival Analysis

Median follow-up was 122 months. During that time, 59 patients died (51 died with disease, eight due to other reasons), 11 are alive with disease, and 12 alive with no signs of disease. One, 2-, 5-, and 10-year OS from the date of diagnosis was 78%, 48%, 30%, and 17%, respectively. Factors that had statistically significant impact on OS in univariate analysis are presented in Table 2. All variables with *p* values of < 0.05 (tumor grade, R0 extent of resection, radical RT, temozolomide-based CTH) were included in the multivariate analysis, which showed that only radical (R0) resection and temozolomide-based CTH are factors independently affecting OS with *p* = 0.005 and *p* = 0.004, respectively (Figure 1a,b, Figure 2).

Table 3 shows factors that had statistically significant impact on PFS in univariate analysis. One, 2-, 5-, and 10-year PFS was 65%, 35%, 17%, and 12%, respectively. Multivariate analysis of factors that had p value below 0.05 in univariate analysis (tumor grade IV, ECOG performance status equal or higher than 2, R0 extent of resection, radical RT) showed that only radical (R0) resection and radical RT are factors independently affecting PFS with *p* = 0.040 and *p* = 0.007, respectively (Figure 1c,d, Figure 2).

Among those who were diagnosed with recurrence of the disease, additional survival analysis was performed. The timing of adjuvant treatment (120 days for RT and 22 days for CTH) was statistically significant in multivariate analysis—with *p* = 0.041 and 0.023, respectively. Treatment of the recurrence and the use of RT in recurrence treatment were also found as statistically significant in terms of survival in this group.

## 4. Discussion

### 4.1. Study Limitations

The limitations of our study are the same as in all of the studies on rare tumors—a retrospective study covering a long time period with different treatment modalities. However, due to the rarity of this tumor in the pediatric population, studies like ours are the only way to collect experience in this topic. Additionally, no unified imaging modality was used for evaluation of surgical outcome, and some patients in early years of the study did not have MR. Furthermore, we did not perform the review of the pathologic diagnosis. Mutations or MGMT methylation are not routinely evaluated in pediatric population and we did not assess its possible impact on survival. What is more, the adjuvant RT and CTH schema varied depending on the institutional protocol at the time of the treatment. Nevertheless, this is one of the biggest reports on HGG, as conducted in seven major Polish hospitals, during the 25 years of the study period.

### 4.2. Clinical Presentation

The diagnosis of HGG, despite improvements in neurosurgery, RT, and systemic therapies, still carries poor prognosis in children and adults. For a long time, a younger age has been believed to be a favorable prognostic factor in HGG, which is in accordance with SEER analyses but in contrast with one report by Finlay J.L. [22,25,33,35]. The impact of age on OS or PFS was not observed in our study. The mean age at the time of disease diagnosis in our group of 12 years old is comparable to the mean age reported by other authors—within the range of 10 to 14 years old [25,33,34]. Looking at the impact of gender on the treatment results, two studies identified male sex as a better prognostic factor, which was in contrast with Finlay J.L.’s observations, but the number of patients included in those analyses was much smaller than in the CCG-945 study [19,22,24]. In accordance with the previous reports and SEER analyses, male sex predilection was observed in our group, but we did not find it statistically significant in terms of OS or PFS [25,33,34]. Prognostic factors for OS found by other authors are presented in Table 4.

Patient’s reported symptoms often are described as signs of increased intracranial pressure, seizures, or paresis, depending on tumor location [1,9,10,14,25]. Clinical presentations of the disease observed in our patients were similar to those described in previous reports. The presence of symptoms such as seizures, hemiparesis, headaches, or others was not found as statistically significant in terms of OS in our group or in the studies of other authors. The duration of symptoms before the diagnosis was described to be shorter for HGG than for lower-grade lesions [36,37]. Roulecke B.C. reported median and average intervals of 24 and 59 days from symptoms onset to the tumor diagnosis with the length being significantly shorter in patients with HGG [36]. This difference was not observed in our group with the same (30 days) median duration of disease symptoms in patients with grade III and grade IV tumors, which is in accordance with the Bilginer B. study, which reported the majority of patients being diagnosed in less than 3 months [14]. In contrast, much longer time intervals with a median of 4 months were reported in other studies [7,9,25,26].

### 4.3. Histopathological Diagnosis

HGG are a histologically heterogeneous group of tumors and are further classified according to the cell of origin as astrocytic tumors (AA, GBM, giant cell GBM, gliosarcoma), oligodendroglial tumors (AO), or oligoastrocytic (mixed) tumors (anaplastic oligoastrocytoma) [38]. Special categories of HGG include diffuse intrinsic pontine gliomas (DIPG); however, such a diagnosis requires a different treatment approach so patients with main tumor location in the brainstem were not included in our analysis. Mathew R.K. reported 5-year OS of 18.9% (United Kingdom, UK) and 30.1% (United States, US) for patients with AA and 8.5% (UK) and 22.3% (US) for GBM [32]. Results observed in our study showed much better OS in AA group (5-year OS of 72%) and comparable OS in GBM patients (5-year OS of 17%). Results observed by other authors are shown in Table 4. Grade IV histopathology was found as statistically significant in terms of OS and PFS in our study in the univariate analysis, as well as in the reports of other authors (Table 4) [6,8,10,19,22,23,28].

### 4.4. Treatment—Surgery

Maximal safe resection is the first step of HHG treatment. Many retrospective series reported extent of resection (EOR) as the only one or one of the main prognostic factors influencing OS (Table 4) [1,6,8,9,10,11,12,13,14,17,19,22,25,27,29,30]. Additionally, SEER analysis performed by Adams H. showed that gross total resection is independently associated with improved OS in GBM patients [34]. Additionally, in our study, the patients who had R0 resection tended to have better OS and PFS compared to those with R1/R2 surgery (*p* = 0.005 and *p* = 0.04, respectively). The Fisher’s exact test was used to determine the association between EOR and grade and showed that EOR is an independent factor influencing survival, irrespective of tumor grade, with *p* = 0.765. These findings are in accordance with the reports of other authors and demonstrate a survival advantage provided by radical resection. Based on those data, an aggressive surgical therapy for malignant pediatric astrocytomas should be considered.

### 4.5. Treatment—Chemotherapy

The current protocol in HGG in children over 3 years old in Poland (which is recommended, but not standard procedure in all Polish hospitals) includes: surgery (maximal safe resection), two to four cycles of CTH (irinotecan + carboplatin), RT (+/− temozolomide), and adjuvant CTH—four cycles of cisplatin/ vincristine/ lomustine and four cycles of temozolomide/cisplatin alternately. However, due to the long study period, varied treatment protocols were employed. Many authors evaluated the role of the systemic treatment and its impact on OS. In the multi-institutional analysis conducted by Walston S., EOR and use of CTH after RT/RCTH were statistically significant for OS and PFS. Furthermore, in patients with subtotal resection/biopsy only, total RT dose of more than 59.4 Gy and adjuvant CTH had a positive impact on OS [11]. Yazici G. found that those who received temozolomide had better OS in GBM group—median OS of 32 vs 12 months [10]. Similar observations were found by Yang T.; however, the difference did not reach statistical significance with *p* = 0.075, and authors did not correlate better survival with the use of this systemic agent but rather with better intraoperative imaging guidance techniques and improved diagnostic imaging [1]. CCG-943 is the only randomized study to date that demonstrated a clear survival advantage associated with adjuvant CTH in children. Patients who were treated with CTH fared significantly better than children treated with RT alone; although, in this trial, unintentional inclusion of low-grade glioma (LGG) as HGG occurred, which resulted in better outcomes [6]. Among other trials, CCG-9933, German HIT-GBM, or French Society of Pediatric Oncology, improvement in OS was shown when adjuvant CTH was implemented before RT, but EOR was still the most important factor affecting survival [15,16,17,18,19,20,21]. Interestingly, in 2015, Mallick S. and Walston S., and in 2018, Azizi A.A., in retrospective analyses, described better OS in group receiving temozolomide [7,8,11]. Their results are in accordance with our observations. Furthermore, better survival observed in the subgroup of patients with disease progression indicate that the delay of over 3 weeks in the beginning of CTH should be omitted. The decision about the therapeutic agent used should be made with caution in patients with grade III tumors, because long administration of temozolomide may lead to mutations in MMR genes and, subsequently, mutations in pathways leading to progression from grade III to grade IV.

### 4.6. Treatment—Radiotherapy

RT as adjuvant treatment (combined with CTH in some patients) was used in almost all of the studies. Doses prescribed were usually within the range of 54 to 60 Gy, given in 1.8 to 2 Gy fractions, and covered tumor (GTV), along with tumor bed and additional margins of 2 to 2.5 cm (CTV,PTV) [7,8,9,10,11,12,14,17,19,22,24,25,28]. Some authors limited the dose to 54 Gy in cases of children 6 years of age or younger, as well as included patients irradiated to only 45 Gy [6,17,27,30]. In the study by Perkins S.M., children received smaller doses (50.4-54 Gy), and in seven patients, RT was delivered to the whole brain. However, her study covered a very long period of time (from 1970 to 2008), which could be the reason for including such patients [13]. Extensive fields and low doses (of 45 to 52.5 Gy) were also used in patients treated in the Sposto R. study, conducted in 1989 [6]. In the Lopez-Aguilar E. study, children underwent hyperfractionated RT to the total dose of 54 Gy, given in two fractions of 1.1 Gy per day, but the authors did not focus on RT outcome and nobody repeated their RT schema [23]. RT as a prognostic factor for better OS was found by Wolff J.E. when doses over 54 Gy were applied [19]. Boudaouara O. found that postoperative RT to the total dose of 60 Gy was significantly associated with better OS [26]. Liu M. found that the use of RT improved OS in patients older than 11 years old [35]. We also observed that the use of radical RT had positive impact on OS and PFS in the univariate analysis, but the multivariate Cox model showed its impact on PFS only. The correlation between the total dose used and the survival was not observed. Given the outcome observed in our patients and the best results observed in studies with RCTH, such adjuvant treatment should be considered in all patients in good performance status after surgery. As shown in the disease progression subgroup, RT should be started sooner than 4 months after the surgery. Furthermore, since 60% of our patients evaluated during systemic treatment showed disease progression, the offset of RT should not be delayed due to extended CTH (except in children younger than 3 years old for whom the decision should be made individually).

### 4.7. Secondary Neoplasms

There are a few inherited syndromes that increase the risk of HGG development, such as Li–Fraumeni syndrome, neurofibromatosis type 1, or Turcot syndrome [26,39]. A family history of brain tumor was present in the cases of five patients in our group. Furthermore, eight patients had the history of previous oncologic treatment—three were previously treated for LGG, two for rhabdomyosarcoma, two for ALL and one for retinoblastoma. Other authors also found that their patients had previous oncologic treatment or tumor-predisposing conditions [40,41]. In Yang T.’s study, five patients received cranial RT during ALL treatment, and three had neurofibromatosis [1]. Additionally, Wolf J.E. described previous neoplasms: one ALL, one ALL and PNET and then HGG, and one craniopharyngioma [17]. LGG were also observed in four patients in Boudaouara O.’s report [26]. Additionally, in Bilginer B.’s study, in two patients, HGG progressed from LGG [14]. The same study reported HGG in patients with: neurofibromatosis type 1, family history of brain tumor, and cranial RT during ALL treatment [14]. Patients with previous history of oncologic treatment, especially those with LGG and ALL, should be considered as high risk for developing secondary high-grade neoplasms, and careful follow-up is recommended [41].

## 5. Conclusions

The treatment outcome of children with HGG is poor. The best results are observed in patients after MR-confirmed R0 resection with immediate postoperative temozolomide-based CTH and RT. Further multi-institutional studies are needed to establish the optimal treatment sequence or more effective treatment regimens.

## Figures and Tables

**Figure 1 cancers-13-02062-f001:**
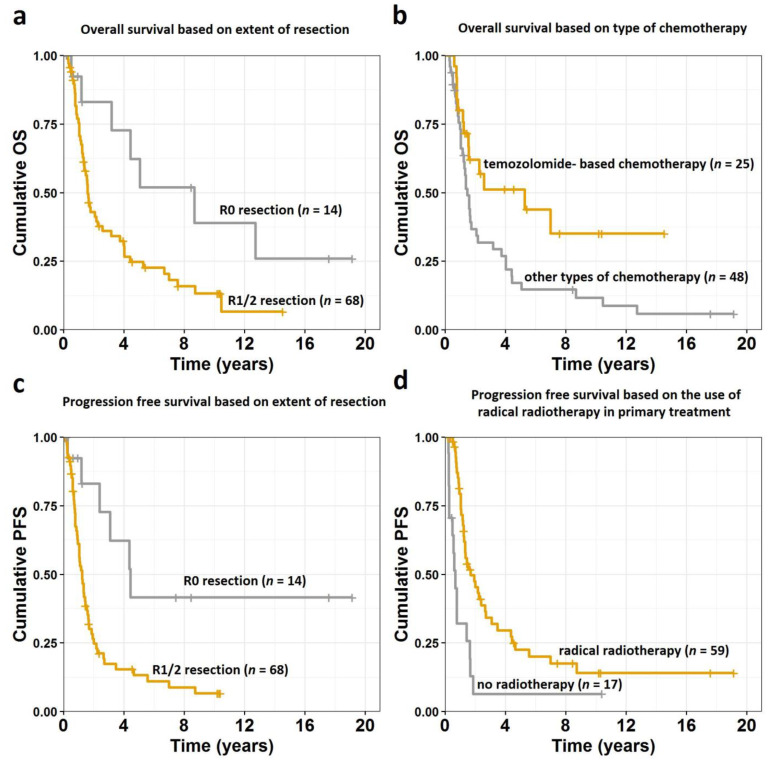
Overall survival curves for all analyzed patients divided into subgroups based on (**a**) extent of resection and (**b**) type of chemotherapy used in primary treatment and progression free survival for all analyzed patients divided into subgroups based on (**c**) extent of resection and (**d**) use of radical radiotherapy in primary treatment. Abbreviations: OS—overall survival; PFS—progression free survival; R0—radical resection (macro- and microscopically); R1—macroscopically radical resection (but not microscopically); R2—macroscopically not-radical resection.

**Figure 2 cancers-13-02062-f002:**
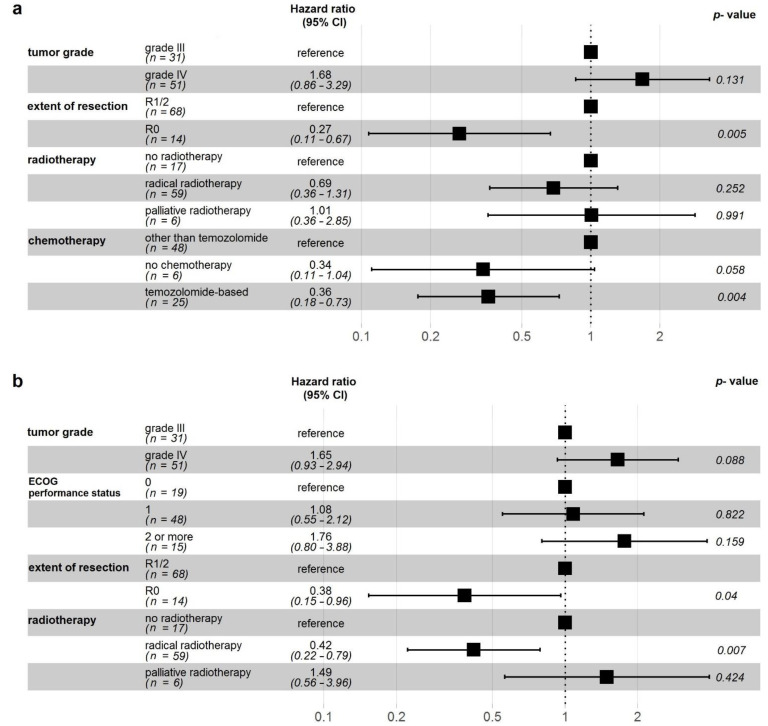
Hazard ratios for overall survival (**a**) and progression free survival (**b**) with 95% confidence intervals and *p*-values calculated from a multivariate Cox proportional-hazards model. Abbreviations: ECOG—Eastern Cooperative Oncology Group; R0—radical resection (macro- and microscopically); R1—macroscopically radical resection (but not microscopically); R2—macroscopically not-radical resection.

**Table 1 cancers-13-02062-t001:** The patients’ and treatment characteristics.

Characteristic		Value
Age at diagnosis	Mean (range)	12 (1–18)

		Number of patients (%)
Sex	Female	35 (42.7)
	Male	47 (57.3)
Pathological subtype	Glioblastoma multiforme G IV	51 (62.2)
	Anaplastic astrocytoma G III	19 (23.2)
	Anaplastic oligodendroglioma G III	12 (14.6)
ECOG performance status	0	19 (23.2)
	1	48 (58.5)
	2	11 (13.4)
	3	3 (3.7)
	4	1 (1.2)
Primary site	Supratentorial	68 (82.9)
	Infratentorial	12 (14.6)
	No data	2 (2.5)
Symptoms	Seizures	17 (20.7)
	Paresis	18 (21.9)
	Headaches	39 (47.7)
	Vomiting/nausea	34 (41.5)
	Visual deficits	9 (11.0)
	Balance disorders	10 (12.2)
	Disturbances of consciousness	16 (19.5)
	Speech problems Hearing deficits	5 (6.1) 4 (4.9)
	Weakness	17 (20.7)
	Other *	29 (35.4)
Underwent surgery	Yes	82 (100)
	R0	14 (17.1)
	R1	16 (19.5)
	R2/biopsy	52 (63.4)
Underwent radiotherapy in primary treatment	Yes	65 (79.3)
	Radical	59
	Palliative	6
	No	17 (20.7)
Underwent chemotherapy in primary treatment	Yes	76 (92.7)
	No	6 (7.3)

Abbreviations: ECOG—Eastern Cooperative Oncology Group; G—grade; R0—radical resection (macro- and microscopically); R1—macroscopically radical resection (but not microscopically); R2—macroscopically not-radical resection. * Other observed symptoms were (the number of symptoms does not reflect the number of patients, as one patient can present with more than one symptom): facial nerve compression symptoms (eight patients), behavior change (six patients), impaired concentration (five patients), somnolence (two patients), memory impairment (two patients), psychotic symptoms (two patients), nervous tics (one patient), numbness of the foot (one patient), dizziness (one patient), aphasia (one patient), forced position of the head (one patient), breathing difficulties (one patient), taste disturbances (one patient), paleness (one patient), asymmetrical positive Babinski symptom (one patient), and tachycardia (one patient).

**Table 2 cancers-13-02062-t002:** Prognostic factors: univariate analysis of overall survival for the whole cohort.

Variable		HR	95% CI	*p*-Value
Age (years)		1	0.96–1.05	0.906
Sex	male	ref.		
	female	0.95	0.57–1.60	0.855
Tumor grade	G III	ref.		
	G IV	2.32	1.31–4.11	**0.004**
ECOG performance status	0	ref.		
	1	1.36	0.73–2.56	0.332
	2 or more	1.76	0.83–3.73	0.138
Tumor location	infratentorial	ref.		
	supratentorial	0.87	0.44–1.72	0.679
Extent of resection	R1/2	ref.		
	R0	0.41	0.18–0.92	**0.030**
Radiotherapy	no radiotherapy	ref.		
	radical radiotherapy	0.52	0.29–0.96	**0.036**
	palliative radiotherapy	1.12	0.41–3.11	0.821
Total dose over 54 Gy	no	ref.		
	yes	0.92	0.50–1.70	0.790
Chemotherapy	other than temozolomide	ref.		
	temozolomide-based	0.49	0.26–0.91	**0.023**
	no chemotherapy	0.46	0.16–1.30	0.144

Abbreviations: ECOG—Eastern Cooperative Oncology Group; G—grade; R0—radical resection (macro- and microscopically); R1—macroscopically radical resection (but not microscopically); R2—macroscopically not-radical resection. The *p*-value of statistically significant (below 0.05) factors were marked in bold to make it more visible for readers.

**Table 3 cancers-13-02062-t003:** Prognostic factors: univariate analysis of progression-free survival for the whole cohort.

Variable		HR	95% CI	*p*-Value
Age (years)		0.99	0.95–1.04	0.706
Sex	male	ref.		
	female	1.06	0.64–1.75	0.822
Tumor grade	G III	ref.		
	G IV	1.80	1.05–3.12	**0.032**
ECOG performance status	0	ref.		
	1	1.72	0.92–3.23	0.091
	2 or more	2.24	1.05–4.79	**0.037**
Tumor location	infratentorial	ref.		
	supratentorial	1.11	0.55–2.26	0.773
Extent of resection	R1/2	ref.		
	R0	0.31	0.13–0.72	**0.006**
Radiotherapy	no radiotherapy	ref.		
	radical radiotherapy	0.36	0.20–0.66	**<0.001**
	palliative radiotherapy	1.75	0.67– 4.57	0.253
Total dose over 54 Gy	no	ref.		
	yes	0.84	0.46–1.52	0.558
Chemotherapy	other than temozolomide	ref.		
	temozolomide-based	0.72	0.41–1.26	0.247
	no chemotherapy	0.51	0.18–1.45	0.210

Abbreviations: ECOG—Eastern Cooperative Oncology Group; G—grade; R0—radical resection (macro- and microscopically); R1—macroscopically radical resection (but not microscopically); R2—macroscopically not-radical resection. The *p*-value of statistically significant (below 0.05) factors were marked in bold to make it more visible for readers.

**Table 4 cancers-13-02062-t004:** Pediatric high-grade gliomas-prognostic factors.

Author (Year of Publication) [Reference Number]	Number of Patients	Treatment Schema	OS	Prognostic Factors for OS
Sposto R. et al. (1989) [6]	58 pts 18 G III 40 G IV	S + RT + CTH or RT alone	1,2,5-y OS G III 65%, 60%, 60% 1,2,5-y OS G IV 65%, 35%, 15%	AA histology, lack of necrosis, location outside basal ganglia, EOR, CTH
Finlay J.L. et al. (1995) [22]	172 pts 82 G III 57 G IV 33 other	S + RT + CTH or CTH-RT-CTH	1,2,5-y OS 80%, 50%, 40%	White race, older age, EOR, AA histology, female sex
Wolf J.E. et al. (2002) [19]	52 pts 25 G III 27 G IV	S + RT + CTH or CTH + RT + CTH (sandwich)	1,2,5-y OS G III 100%, 100%, 100% 1,2,5-y OS G IV 100%, 45%, 40%	EOR, male sex, AA histology, residual tumor > 2 cm, RT ≥ 54 Gy, sandwich CTH
López-Aguilar E. et al. (2003) [23]	25 pts 20 G III 5 G IV	S + CTH + hyperRT + CTH	1,2,5-y OS 67%, 67%, 67%	AA histology, supratentorial location
Wolf J.E. et al. (2010) [17]	97 pts 30 G III 41 G IV 26 no data	S + RT + CTH	1,2,5-y OS 56%, 30%, 19%	EOR
Perkins M. et al. (2011) [13]	24 pts G IV	S + RT +/− CTH	1-2-5-y OS 57%, 32%, 10%	EOR
Cohen K.J. et al. (2011) [30]	90 pts 31 G III 55 G IV 4 other	S + RCTH +CTH	1,2,5-y OS 70%, 35%, 15%	EOR
Das K.K. et al. (2012) [25]	65 pts G IV	S + RT + CTH	1,2,5-y OS 55%, 30%, 15%	EOR
Yang T et al. (2013) [1]	37 pts G IV	S + RT + CTH	1,2,5-y OS 64%, 45%, 18%	EOR
Cabanas R. et al. (2013) [31]	23 pts 19 G III 4 G IV	S + RT + CTH + nimotuzumab	1,2,5-y OS 64%, 54%, not reported	Not reported
Walston S. et al. (2015) [11]	51 pts, 23 G III 28 G IV	S +/- RT/RCTH +/− CTH	1,2,5-y OS 87%, 59%, 37% 1,2,5-y OS GIII 90%, 75%, 50% 1,2,5-y OS GIV 90%, 30%, 20%	EOR, CTH after RT/RCTH
Jung T.-J. et al. (2015) [27]	62 pts 28 G III 34 G IV	S + RCTH or RT+CTH or RT	1,2,5-y OS 76%, 35%, not reported	EOR, cerebrospinal fluid dissemination
Mallick S. et al. (2015) [7]	23 pts G IV	S + RCTH +CTH	1,2,5-y OS 70%, 61%, 40%	CTH-TMX
Yazici G. et al. (2016) [10]	63 pts, 26 G III 37 G IV	S + RT +/− CTH	1,2,5-y OS G III 79%, 63%, 35% 1,2,5-y OS G IV 48%, 30%, 22%	AA histology, better PS, EOR, CTH-TMX
Espinoza J.C. et al. (2016) [24]	32 pts 19 G III 11 G IV 2 other	S + CTH + AuHPCR +/− RT (50%)	1,2,5-y OS 70%, 50%, 36%	Male sex
Jalali R. et al. (2016) [12]	66 pts G IV	S + RCTH +CTH	1,2,5-y OS 62%, 30%, 20%	EOR, thalamic location
Gupta S. et al. (2017) [9]	51 pts G IV	S + RCTH +CTH	1,2,5-y OS 60%, 40%, 20%	EOR, CTH-TMX
Bilginer B. et al. (2017) [14]	42 pts G IV	S + RT + CTH	1,2,5-y OS 50%, 19%, 10%	EOR, lack of seeding metastases
Lucas J.T. et al. (2017) [28]	56 pts 29 G III 27 G IV	S + RT + erlotynib	1,2,5-y OS 45%, 25%, 20%	AA histology, lack of brainstem/cerebellar involvement
Lee J.W. et al. (2017) [29]	30 pts 7 G III 16 G IV 7 other	S + RT or CTH +/− HDCT/auto-SCT	3-y OS 32%	EOR
Azizi AA. et al. (2018) [8]	38 pts 17 G III 21 G IV	S + RCTH +CTH	1,2,5-y OS 68%, 29%, 21%	EOR, lower grade, CTH-TMX
Boudaouara O. et al. (2019) [26]	30 pts 3 G III 27 G IV	S + RT +/− CTH	Median OS 13 months; 1,2,5-y OS described only in subgroups	Postoperative RT, complete response

AA—anaplastic astrocytoma; AuHPCR—autologous hematopoietic progenitor cell rescue; CTH—chemotherapy; EOR—extent of resection; G—grade, HDCT/auto-SCT—high-dose chemotherapy and autologous stem cell transplantation; hyperRT—hyperfractionated radiotherapy; OS—overall survival; PS—performance status; pts—patients; RCTH—radiochemotherapy; RT—radiotherapy; S—surgery; TMX—temozolomide; y—year.

## Data Availability

The data presented in this study are available in this article.

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
