# Peer review of "High-Grade Gliomas in Children—A Multi-Institutional Polish Study"

_cancers, 2021, doi:10.3390/cancers13092062_

Round 1
Reviewer 1 Report
This is a very well written manuscript, describing a thorough analysis of a retrospective patient dataset and placing it in context with the findings of other patient studies.
I have a small number of suggestions which I believe will further improve the manuscript:-
The size of the dataset and the time scale that it encompasses presents some further limitations to the analyses and these must be acknowledged throughout rather than acknowledged in a paragraph at the end.
How many patients are represented in each of the curves shown in Figure 1. Please give n=? in each case.
Only 30% of patients received TMZ. Were the demographics of the patients that received TMZ the same as the trial as a whole e.g. mainly grade IV? Secondly, it is well known that TMZ is mutagenic if MMR genes are mutated. Continuous administration of TMZ may lead to mutations in MMR genes, and subsequently mutations in pathways leading to progression from III to IV. I realise that that we need to improve survival rates and this is one of the few effective options but these risks and their significance over the lifetime of a child should be acknowledged.
Please state whether EOR is independent of grade.
Author Response
Response to the Reviewers
We would like to thank the editor and the reviewers for careful and thorough reading of this manuscript and for the thoughtful comments and constructive suggestions, which help to improve the quality of this manuscript. Our response follows (the reviewers comments are in a bold black font, changes in the manuscript are in a red font).
The Reviewer 1 wrote:
This is a very well written manuscript, describing a thorough analysis of a retrospective patient dataset and placing it in context with the findings of other patient studies. I have a small number of suggestions which I believe will further improve the manuscript:
The size of the dataset and the time scale that it encompasses presents some further limitations to the analyses and these must be acknowledged throughout rather than acknowledged in a paragraph at the end.
Thank you very much for your suggestion. The number of patients included in our study (82) is comparable to the number of patients included in reports of other authors. As mentioned in Table 4 only 3 of 22 cited studies included more patients than we did. We would rather see that as one of the strengths of the analysis instead of limitations. In such rare diseases only time scale and multi-center cooperation can provide inclusion of larger number of patients. Of course we are aware that the longer period of time covered the more treatment schema could change and this is one of the main limitations of this study. The period of time covered in this study is mentioned in section Material and Methods on page 2, line 80. As suggested by the Reviewer the section ‘Study limitations’ was moved at the beginning of Chapter 4. Discussion on page 10, lines 265-276 and marked with red font.
How many patients are represented in each of the curves shown in Figure 1. Please give n=? in each case.
The number of patients represented in each of the curves shown in the Figure 1 is as follows:
1a: R0 – n=14 patients, R 1/2 – n=68 patients
1b: temozolomide-based chemotherapy – n=25 patients, other types of chemotherapy – n=48
1c: R0 – n=14 patients, R 1/2 – n=68 patients
1d: radical radiotherapy – n=59 patients, no radiotherapy – n=17 patients
The numbers has been added to all the curves on the Figure 1 as suggested by the Reviewer (page 7, line 240).
Only 30% of patients received TMZ. Were the demographics of the patients that received TMZ the same as the trial as a whole e.g. mainly grade IV? Secondly, it is well known that TMZ is mutagenic if MMR genes are mutated. Continuous administration of TMZ may lead to mutations in MMR genes, and subsequently mutations in pathways leading to progression from III to IV. I realise that that we need to improve survival rates and this is one of the few effective options but these risks and their significance over the lifetime of a child should be acknowledged.
Firstly, 12 out of 25 patients who received TMZ had WHO G III tumors and 13 had WHO G IV tumors. The proportions (48% G III and 52% G IV) are slightly different than in the whole group in which Grade IV tumors constitute 62% and Grade III 38% of all tumors. As suggested by the Reviewer this information has been added to the manuscript to the section Results on page 4 lines 152-153 and marked with red font.
Secondly, as suggested by the Reviewer the information about mutagenic side effects of TMZ has been added to the section Discussion on page 13, lines 367-371 and marked with red font: ‘The decision about the therapeutic agent used should be made with cautious in patients with grade III tumours because long administration of temozolomide may lead to mutations in MMR genes and subsequently mutations in pathways leading to progression from grade III to grade IV.’
Please state whether EOR is independent of grade.
Extent of the resection (EOR) was evaluated in all the cases and defined as: R0 – radical resection (macro- and microscopically); R1 – macroscopically radical resection (but not microscopically); R2 – macroscopically not-radical resection. R0 resection confirmed with MR was performed in 14 patients. Among them 6 had Grade III tumors and 8 had Grade IV tumors. As suggested by the Reviewer this information has been added to the manuscript to the section Results on page 4 lines 145-146 and marked with red font. Additionally, the Fisher’s exact test was used to determine the association between EOR and grade and showed that EOR is an independent factor influencing survival irrespective of tumor grade with p=0.765 (see Table below).
Calculations:
|
grade/ EOR |
RO |
R1/2 |
|
III |
6 (42,9%) |
25 (36,8%) |
|
IV |
8 (57,1%) |
43 (63,2%) |
p- value= 0.765
The information about the Fisher’s exact test was added to the section Material and Methods on page 3 lines 111-112 and marked with red font: ‘the Fisher’s exact test was used to determine the association between EOR and grade’. Information about the results of the analysis was added to section Discussion on page 13, lines 334-336 and marked with red font: ‘The Fisher’s exact test was used to determine the association between EOR and grade and showed that EOR is an independent factor influencing survival irrespective of tumor grade with p=0.765.’
We are aware that tumor grade has been described by other authors as independent prognostic factor and that univariate analysis of our group showed that tendency, although we could not make such conclusions based on the results of multivariate analysis. Nevertheless, as showed on Figure 2 patients with Grade IV tumors tended to have worse overall and progression free survival.
Reviewer 2 Report
You collected patient data concerning pediatric high grade gliomas treated in 7 different clinics, using different treatment regimens and diagnostic tools and desrcibed your data in a proper fashion.
In Table I R1 is explained in he legenda as microscopic but not macroscopic total resection? This appears contradictory.
Your main message early Radiotherpay and Temozolomide preferably in combination with gross total resection is not new, but you provide data form another retrospective series.
Your remarks in de discussion about the shortcomings of this study are convincing and realistic.
It is unfortunate that relevant diagnostic data like MGMT status, molecular genetic characterisitics and functional outcome data could not be collected in this series.
Your series illustrates that more effective treatment regimens should be developed for this patient group.
Author Response
Response to the Reviewers
We would like to thank the editor and the reviewers for careful and thorough reading of this manuscript and for the thoughtful comments and constructive suggestions, which help to improve the quality of this manuscript. Our response follows (the reviewers comments are in a bold black font, changes in the manuscript are in a red font).
The Reviewer 2 wrote:
You collected patient data concerning pediatric high grade gliomas treated in 7 different clinics, using different treatment regimens and diagnostic tools and described your data in a proper fashion.
In Table I R1 is explained in the legenda as microscopic but not macroscopic total resection? This appears contradictory.
Yes, of course we agree with the Reviewer – R1 resection is macroscopically radical resection (but not microscopically). A typo was made during the manuscript preparation. We corrected that in all the legends within the article (page 4, lines 129-130; page 7, lines 230-231; page 8, lines 246-247; page 8, lines 250-251; page 9, line 256).
Your main message early Radiotherapy and Temozolomide preferably in combination with gross total resection is not new, but you provide data form another retrospective series.
Not all of the studies on paediatric patients showed that temozolomide is an effective therapeutic agent in that group of patients. As showed in Table 4 temozolomide-base chemotherapy was found as a positive prognostic factor only in 4 studies of Azizi AA, Gupta S, Yazici G and Mallick S. The first systemic agents used in this population and described in 1989 by Sposto R were nitrosourea, vincristine and prednisone, and temozolomide-base chemotherapy was introduced in the last ten years. What is more, while the median time to the start of systemic treatment in our group was only 30 days, the median time to the beginning of radiotherapy was 132 days. It is worth mentioning, that in reports concerning adult patients treatment results shorter time to radiotherapy onset had positive impact on survival. What is more, the sub-analysis of the group with recurrence patients suggested that those who received irradiation within the first four months after the surgery had better overall survival. Such observation was not described before and the one study addressing the radiotherapy timing issue by Azizi AA at al. ‘Does the interval from tumour surgery to radiotherapy influence survival in paediatric high grade glioma?’ did not found such correlation. All mentioned above makes our study a valuable input into the research on paediatric patients with high grade glioma.
Your remarks in the discussion about the shortcomings of this study are convincing and realistic. It is unfortunate that relevant diagnostic data like MGMT status, molecular genetic characteristics and functional outcome data could not be collected in this series.
We are aware that those are the limitations of our study. MGMT status and molecular genetic characteristics are not routinely evaluated in paediatric patients in Poland. Such data collection would require re-evaluation of tissues obtained during the surgery what could be impossible in case of patients treated at the beginning of the study (the time of such material storage according to the national law is 20 years) and very hard in all other patients as regional neurosurgical departments were not included into participation in this study. The evaluation of functional outcome of paediatric patients who received treatment due to brain tumor would be an interesting topic, although we did not collected such data. Routine follow-up in paediatric patients ends at the age of 18 years old and the cooperation between paediatric and adult oncologic centres is the issue of debate and development. This together makes such data hard to obtain. Nevertheless, we would keep that comment in mind for our future studies.
Your series illustrates that more effective treatment regimens should be developed for this patient group.
We agree with the Reviewer opinion that current treatment regiments in paediatric HGG are not producing good results. We changed the Conclusions as suggested by the Reviewer (page 14, lines 420-421): ‘Further multi-institutional studies are needed to establish the optimal treatment sequence or more effective treatment regimens.’ and marked the change with red font.
Reviewer 3 Report
The authors focus on an interesting topic of high grade glioma (HGG) in children. The manuscript described a retrospective analysis of 83 patients treated for HGG. The study investigated the long-term results of treatment of children with HGG and to identified factores related to better survival. However the authors need to adress some major issues in this manuscript.
There are some suggestions and comments:
- Although high grade glioma in children is a relatively rare tumor., it is still very difficult to treat and there is no consensus on how to treat it. This study is attractive.
- Line 119. 8 patients had another tumors, did the include in this study or not? Please clarify the included and excluded criteria.
- There is an item of "Other" in Table one. Please specify under the table, what are the "others" that are included.
- In this manuscript, there is no "Stupp's protocol":RT?TMZ+TMZ alone- treatment group? What is the reason?
Author Response
Response to the Reviewers
We would like to thank the editor and the reviewers for careful and thorough reading of this manuscript and for the thoughtful comments and constructive suggestions, which help to improve the quality of this manuscript. Our response follows (the reviewers comments are in a bold black font, changes in the manuscript are in a red font).
The Reviewer 3 wrote:
The authors focus on an interesting topic of high grade glioma (HGG) in children. The manuscript described a retrospective analysis of 83 patients treated for HGG. The study investigated the long-term results of treatment of children with HGG and to identified factors related to better survival. However the authors need to address some major issues in this manuscript.
There are some suggestions and comments:
- Although high grade glioma in children is a relatively rare tumor., it is still very difficult to treat and there is no consensus on how to treat it. This study is attractive.
- Line 119. 8 patients had another tumors, did the include in this study or not? Please clarify the included and excluded criteria.
The presence of previous neoplasm was not an exclusion criterion. As suggested by the Reviewer, to clarify this issue the phrase ‘The presence of previous neoplasm was not an exclusion criterion.’ was added to Material and Methods section on page 2, lines 87-88 and marked with red font.
- There is an item of "Other" in Table one. Please specify under the table, what are the "others" that are included.
Table 1 addresses the patients’ and treatment characteristics. ‘Other’ mentioned by the Reviewer refers to symptoms which were present in patients before tumor diagnosis. The number and the variety of those symptoms prevented us from initial inclusion into the article.
Other observed symptoms were (the number of symptoms does not reflect the number of patients as one patient can present with more than one symptom): facial nerve compression symptoms (8 patients), behaviour changes (6 patients), impaired concentration (5 patients), somnolence (2 patients), memory impairment (2 patients), psychotic symptoms (2 patients), nervous tics (1 patient), numbness of the foot (1 patient), dizziness (1 patient), aphasia (1 patient), forced position of the head (1 patient), breathing difficulties (1 patient), taste disturbances (1 patient), paleness (1 patient), asymmetrical positive Babinski symptom (1 patient), tachycardia (1 patient).
As suggested by the Reviewer this information (as mentioned above) was added under the Table 1 (page 4, lines 131-135) and marked with red font.
- In this manuscript, there is no "Stupp's protocol":RT?TMZ+TMZ alone- treatment group? What is the reason?
Stupp protocol is not routinely applied in paediatric population in Poland. As mentioned in the Discussion on page 13, lines 338-32: ‘The current protocol in HGG in children over 3 years old in Poland (which is recommended, but not standard procedure in all polish hospitals) includes: surgery (maximal safe resection), 2 to 4 cycles of CTH (irinotecan + carboplatin), RT (+/- temozolomide) and adjuvant CTH – 4 cycles of cisplatin/ vincristine/ lomustine and 4 cycles of temozolomide/cisplatin alternately’ typical treatment in paediatric population differs from the one implemented in adult patients.
Round 2
Reviewer 2 Report
Dear authors,
Thank you for addressing the previous comments and for improving the text.
Your paper describes relevant retrospective data concerning the treatment of HGG in children, and can be valuable for designing future clinical trials.